# The Impact of Controlled Fermentation Temperature on Chemical Composition and Sensory Properties of Cacao

**DOI:** 10.3390/foods14091441

**Published:** 2025-04-22

**Authors:** Ana M. Calvo, Andrea C. Montenegro, Diana M. Monroy, Lucero G. Rodriguez-Silva, Ariel R. Carreño-Olejua, Ivan D. Camargo

**Affiliations:** 1Corporación Colombiana de Investigación Agropecuaria-AGROSAVIA, Centro de Investigación Tibaitatá–Km 14 Vía Mosquera, Mosquera 250047, Cundinamarca, Colombia; acalvo@agrosavia.co; 2Corporación Colombiana de Investigación Agropecuaria-AGROSAVIA, Centro de Investigación Caribia–Km 6 Vía Sevilla-Guacamayal, Zona Bananera 502041, Magdalena, Colombia; dmmonroy@agrosavia.co; 3Corporación Colombiana de Investigación Agropecuaria-AGROSAVIA, Centro de Investigación La Suiza–Km, 32 Vía al mar, Puerto Arturo, Rionegro 687527, Santander, Colombia; lgrodriguez@agrosavia.co (L.G.R.-S.); acarreno@agrosavia.co (A.R.C.-O.); idcamargo@agrosavia.co (I.D.C.)

**Keywords:** phenolic substances, UPLC-DAD-RI, *Theobroma cacao* L., bitterness, astringent, temperature

## Abstract

The content of phenolic compounds can affect the quality of cacao beans (*Theobroma cacao* L). The variation in the concentration of these compounds is influenced by factors such as cacao variety, fermentation conditions, and temperature, which play a crucial role in the method of bean drying. In this study, the analytical method of ultra-performance liquid chromatography (UPLC-DAD-RI) was developed to identify, quantify, and examine variations in the concentrations of catechins (catechin, epicatechin, and epigallocatechin) and methylxanthines (theobromine and caffeine) by subjecting the beans to controlled temperature fermentation. Three temperature-controlled treatments were used during fermentation on three cacao genotypes (CCN 51, ICS 95, and TCS 01). The average temperature in different treatments was T1: 41.14 ± 3.84 °C, T2: 42.43 ± 4.39 °C, and T3: 43.86 ± 4.74 °C. The results demonstrate variations in the concentration of phenolic compounds across the evaluated treatments (T1, T2, and T3). Catechin levels rose from the beginning of fermentation up to day 5, after which they declined by day 6. Conversely, theobromine and caffeine concentrations decreased until day 5, then increased by day 6. A sensory analysis revealed a basic flavor profile (bitter, astringent, and acidic) that was balanced by enhancements in specific attributes, highlighting fruity, citrus, and cacao notes. A significant correlation (*p* < 0.05) was found between bitterness and the concentrations of epigallocatechin, caffeine, epicatechin, and total phenols. In contrast, a low correlation was observed between bitterness and theobromine and catechin. The astringent profile was directly correlated with epigallocatechin concentration and moderately correlated with theobromine and catechin levels. Acidic flavors showed a moderate correlation with epigallocatechin concentration. The cacao flavor was correlated with catechin and total phenols, while the citrus flavor was linked to total phenol concentration. Notably, the decrease in phenolic compound concentrations and sensory analysis suggested that the higher fermentation temperatures observed in T3 may enhance the development of a superior flavor quality in cacao.

## 1. Introduction

Proper post-harvest processing of cacao, including fermentation, can affect the quality of cacao beans. It can substantially increase market value when aligned with industry and market standards [1]. The variety of the bean, the geographical origin, the stage of maturity, and fermentation and drying techniques throughout the production and processing cycle are various factors that influence the chemical composition of cacao beans and their byproducts [2]. The degradation of polyphenols, which actively influence color changes and the bitter and astringent taste of the beans, is among the most critical transformations during cacao fermentation [3].

Due to their antioxidant properties, polyphenols exhibit various biological activities, including anti-cancer, anti-inflammatory, and anti-diabetic effects [3]. These compounds are characterized by having more than one phenolic group and are defined by an aromatic ring containing one or more hydroxyl groups. Esters, methyl esters, glycosides, and others are functional derivatives of polyphenols [4]. These compounds are present in plant tissues; they are often bound to sugars (glucose, galactose, and glucuronic acid) or associated with carboxylic acids, lipids, and other phenolic compounds [5].

Additionally, they contribute a significant role in the sensory quality of plant-based foods. For instance, anthocyanins and anthocyanidins are responsible for red, blue, and violet pigmentation in fruits and vegetables and exhibit strong antioxidant properties [4]. Condensed tannins (proanthocyanidins), another group of polyphenols, are associated with astringency in certain foods [5,6].

The most abundant polyphenols in cacao beans are flavonoids, primarily catechins (37%), proanthocyanidins (58%), and anthocyanins (4%). Epicatechin constitutes approximately 30% of the total polyphenol content in cacao beans [7]. Fermentation significantly affects the flavonoid content, contributing to the astringency and color of beans [8,9]. Furthermore, these compounds are ethanol-soluble, allowing for easy extraction [10]. Notably, epicatechin levels increase during ripening, while catechin levels remain stable [3].

Methylxanthines—specifically theobromine, caffeine, and theophylline—are alkaloids responsible for cacao’s bitter and astringent taste [11]. These compounds, which also provide psychostimulant health benefits, vary with bean type, plant variety, and fermentation level [12]. Although methylxanthines do not undergo chemical transformations during fermentation, approximately 30% are lost due to diffusion and migration outside the bean [13].

Polyphenols diffuse from cellular compartments and undergo oxidation, giving rise to high-molecular-weight insoluble tannins during fermentation. Polyphenol oxidase catalyzes these reactions but is largely inactivated within the first day of fermentation, reducing enzymatic activity from 50% to 6% by the second day [12]. Temperature plays a critical role in fermentation, influencing the formation of desirable flavor compounds and reducing undesirable ones [14,15]. For example, an isothermal range of 45–50 °C promotes flavonoid degradation, thereby reducing bitterness and astringency [16]. It should be noted that a complex mixture of flavonoids, alkaloids (such as theobromine) and other polyphenols also contributes to cocoa’s bitter taste [17].

Recent findings suggest that thermostatically controlled fermentation, maintaining temperatures between 45–48 °C from days four to six, enhances the development of fine-flavor cacao notes [15]. While previous research on spontaneous fermentation has documented the impact of temperature fluctuations on phenolic compound transformations [2,15,18], studies on controlled fermentation conditions are scarce. An important research gap is the lack of comprehensive data on how temperature control influences the retention or degradation of key phenolic compounds (catechins and methylxanthines). This study addresses this gap by systematically evaluating the effect of controlled fermentation temperature on the concentration of catechins and methylxanthines and sensory attributes in cacao beans. Investigating how temperature modulates these chemical constituents provides deeper insights into the relationship between fermentation conditions, cocoa quality, and flavor development, ultimately contributing to improved post-harvest processing strategies for fine-flavored cocoa production.

## 2. Materials and Methods

### 2.1. Chemicals and Reagents

The standards for (+)-catechin, (-)-epicatechin, theobromine, (-)-epigallocatechin, and caffeine were sourced from Sigma Aldrich (St. Louis, MO, USA). A stock solution of 1 mg/mL was prepared by dissolving (-)-epicatechin, (+)-catechin, (-)-epigallocatechin, and caffeine in acetonitrile. In contrast, theobromine was dissolved in a mixture of acetone, Milli-Q water, and acetic acid (70:29.5:0.5, *v*/*v*/*v*). The working standard solutions were prepared from the stock solutions in extraction solvent A (acetone/water/acetic acid, 70:29.5:0.5, *v*/*v*/*v*). Methanol, acetonitrile, and acetone used in the study were of HPLC grade (Merck, Darmstadt, Germany), and ultrapure water was obtained from Milli-Q HX 7040^®^ equipment (Merck Millipore, Burlington, NJ, USA).

We used Sigma-Aldrich^®^-brand standards of Tannic Acid with purity of ≥99.0; Folin–Ciocalteu’s 2N Phenol Folin-Ciocalteu’s

Reagent (Sigma-Aldrich, St. Louis, MO, USA); and anhydrous sodium carbonate (NaCO_3_) ≥ 99.9% (Supelco, Burlington, USA).

All solutions were prepared on the same day as the determinations, protected from direct sunlight, and stored at 4 °C. However, to maintain the same conditions in the sample extracts, a mixture of methanol, water, and formic acid (70:29.5:0.5, *v*/*v*/*v*) was used as the solvent, and the solution was further diluted at a 1:9 ratio with the same mixture.

### 2.2. Apparatus

All samples were dried using a DIN 40050-IP20 recirculation oven (Memmert, Schwabach, Germany) and ground using a coffee grinder (KitchenAid, Greenville, NC, USA). UPLC analyses were performed using the UPLC-DAD-RI system (Acquity 2010 model, Waters Inc., Milford, MA, USA). The UPLC analyses were conducted using an Acquity HSS T3 column (100 mm × 2.1 mm, 1.8 µm particle size) (Waters, Milford, MA, USA) with a binary mobile phase at a flow rate of 0.4 mL/min. The mobile phase consisted of eluent A (water/formic acid, 99.9/0.1, *v*/*v*) and eluent B (acetonitrile). The elution gradient at 30 °C was set as follows: 0–10 min, 98% A and 2% B.

For the UV–Vis method, a 96-well plate reader was used, with a wavelength range of 200 to 999 nm and measurement range of 0.000 to 4.000 OD (Biotek, Winooski, VT, USA), a high-speed centrifuge, digital control, maximum speed 20,000 rpm, temperature control: refrigerated Z32 HK (Hermle Benchmark, Sayreville, NJ, USA) and a microcentrifuge maximum capacity × 1.5–2 mL, maximum speed 18,000/31,514 rpm/RCF, and temperature control −20 to + 40 °C Micro 220R Zoor (Hettich, Tuttlingen Germany). A concentrator (Eppendorf, Concentrator plus/Vacufuge^®^ plus) and a fat extraction system FOSS, SoxtecTM 2050 (FOSS, Hilleroed, Denmark) were used.

### 2.3. Collection and Preparation of Samples

Samples of cacao beans from three genotypes—TCS01, ICS95, and CCN51—were collected in San Vicente de Chucurí, Santander, Colombia. The bean samples used in this study were the same materials analyzed in a previous study by [15]. The fermentation conditions and sampling schedule were also consistent with the methodology employed in the earlier research. 

Fermentation was conducted at the La Suiza Research Center (Agrosavia) in Rionegro, Santander, Colombia (7°22′13″ N, 73°10′39″ W) using a Memmert IN450 incubator. The study aimed to evaluate the impact of controlled temperature on the concentration of phenolic compounds for three different temperature profiles (treatments) applied to the three cacao genotypes, as detailed in Table 1.

The treatments were mainly different by the profile of temperature used throughout fermentation time [15]; little mean differences in the profile potentially might lead to very different outcomes in the sensory profiles [16]. Thus, the temperature profile used had two different phases: an exothermic (almost a monotonic increase in temperature during the first four days) and an isothermal phase (from four to six days). The exothermic phase, with slight differences in the temperature increase between treatments, was settled to achieve an isothermal phase that maintained the differences between treatments at high temperatures (Table 1) (see details in Supplementary Information of Camargo et al., 2024 [15]).

According to [15], fermentation experiments were carried out in stainless steel vessels featuring perforations across their surface. These perforations enhanced the drainage of fermentation sweat and improved heat transfer within the fermentation mass. To approximate the biochemical changes in polyphenol concentration, grain samples were collected at four points during the fermentation process (0, 48, 120, and 144 h), corresponding to days 0, 2, 5, and 6 of fermentation.

For each genotype, the sampling unit comprised a composite sample taken from two randomly assigned experimental units, one from each collection tray. This resulted in three sampling units per genotype (14, 25, and 36 samples). All samples were collected under strict safety protocols to prevent contamination. Photographs of the method are presented in Appendix A.

The cacao bean mucilage was removed using paper towels, after which the nib and testa tissues were manually separated. These components were then oven-dried at 70 ± 0.1 °C for at least 72 h until they reached a constant weight. Finally, the dried testa and nibs were ground using a coffee grinder to obtain a homogeneous sample.

Degreasing and fat content were determined using the Soxhlet gravimetric method, as proposed in AOAC 963.1564. Hexane was used as the solvent for extraction.

### 2.4. Extraction and Analysis of Cacao Phenolic-Rich Extract

For the extraction of polyphenols the modified method proposed by [19] was followed. First, 0.1 g of dry, ground, and defatted cacao nibs were weighed and mixed with 80 mL of analytical grade hexane for 1.5 h from 2 g of sample in 2 mL Eppendorf tubes. Next, 500 µL of the extraction solution (acetone/MilliQ water/acetic acid, 70/29.5/0.5) was added to each tube, which was then vortexed for 2 min and shaken on an orbital shaker for 5 min. The tubes were centrifuged at 5000 rpm for 5 min at 4 °C, and the supernatant was collected in another microtube. This extraction process was performed in triplicate for each sample.

Subsequently, the combined supernatants of each sample were filtered through glass microfiber discs (Whatman^®^ glass fiber, 1.2 microns) using a small funnel (Boeco, Hamburg, Germany). Acetone was evaporated using a SpeedVac at 30 °C for 2 h. The resulting aqueous phase was diluted with MilliQ water using a dilution factor of 40 and passed through syringe filters with a 0.22 µm PTFE membrane.

#### 2.4.1. Ultra-Performance Liquid Chromatography (UPLC-DAD-RI)

The sample volume injected in the UPLC System was 2.5 µL. The quantification was performed using the calibration curves of epicatechin, catechin, theobromine, epigallocatechin, and caffeine. The results were expressed as mg of phenolic compound per gram of phenolic extract.

#### 2.4.2. Ultraviolet–Visible (UV–Vis) Spectroscopy 

For the determination of total polyphenols, the modified Folin–Ciocalteu method was used. First, 100 mg of dried and ground cacao nib sample was weighed, and the volume of the extraction solution was reduced 4 times to 2.5 mL. The extraction was performed in duplicate to ensure the complete obtainment of total phenols. The extraction solution used was methanol/water/formic acid (70.0:29.5:0.5). A centrifugation step was added at 6000 rpm for 8 min at 4 °C and supernatants were mixed (concentrated extract) and stored in a 15 mL conical tube.

Quantification was performed in 96-well microplates. A dilution of the concentrated extract (1:19) was made, 75 µL of the diluted extract or the standard was added to the well, 38 µL of 1 N Folin–Ciocalteu reagent was added, and 187 µL of 20% sodium carbonate was added. The mixture was incubated at room temperature for 40 min in darkness, and the absorbance was measured at 755 nm using a plate reader (Biotek, Synergy HT).

### 2.5. Method’s Validation

For the validation of the methods by UPLC-DAD-RI and UV–Vis, calibration curves were used, starting with the serial dilution of the stock solution of the standards: catechin, epicatechin, epigallocatechin, caffeine, theobromine, and tannic acid for total phenols. The parameters considered for validation were linearity, limits of detection (LOD), limits of quantification (LOQ), and precision.

#### 2.5.1. Linearity

Linearity was evaluated using calibration curves based on the integrated peak area. These curves were generated from ten concentration points ranging from 0.005 to 50 mg/mL, with each standard solution injected three times. The mean absorbance values of the analytes were plotted against their respective concentrations, and the curves were constructed using the linear least squares regression method.

#### 2.5.2. Limits of Detection (LOD) and Quantification (LOQ)

The LOD and LOQ were determined by analyzing ten independently prepared blank samples. Following the guidelines of the International Union of Pure and Applied Chemistry (IUPAC), the LOD was calculated as three times the standard deviation of the blank measurements, while the LOQ was defined as ten times the standard deviation.

#### 2.5.3. Precision

Repeatability was assessed by analyzing two different concentrations of the pure standard five times within the same day. To evaluate intermediate precision, the experiment was repeated using the same concentrations on different days. The relative standard deviation (RSD) was calculated for both intra-day and inter-day measurements to assess the consistency of the analytical method.

### 2.6. Method’s Application

#### Phenolic Compounds of Cacao Nibs with Temperature-Controlled Fermentation

To apply the polyphenol quantification process in cacao using UPLC-DAD-RI and UV–Vis, 108 samples of cacao beans were analyzed at different stages of the fermentation process. This analysis was conducted on three cacao varieties (CCN51, ICS95, and TCS01), each subjected to three fermentation temperature profiles corresponding to treatments T1, T2, and T3 (Table 1). These profiles were designed to increase the average fermentation temperature, aiming to analyze its effect on phenolic compound content.

### 2.7. Sensory Analysis

Sensory analysis was conducted on 54 samples of cacao liquor, considering three genotypes, three fermentation treatments, and two evaluation time points (day five and day six) of fermentation, as reported by [15] to relate the sensory profile to the total polyphenol concentration. In summary, cocoa nibs were obtained through roasting, shelling, and grinding to liquefy the cacao liquor at 55 ± 0.1 °C [15]. A net amount of 5 ± 0.01 g of liquor was provided in duplicate to eleven trained evaluators and one expert leader of the CI La Suiza sensory panel who carried out a quantitative analysis on aroma and flavor descriptors in 18 tasting sessions, with six liquors assessed per session (see Camargo et al., 2024 for details [15]). The sensory evaluation included basic flavors (astringency, bitterness, and acidity), specific attributes (cocoa, fruity, floral, nutty, sweet, and herbal), and acquired defects: over fermented, moldy, damp, and rancid. Flavor attributes were rated on a scale from 0 to 10, where 0 indicated absence and 10 represented high intensity [15].

### 2.8. Statistical Analysis

The association between phenolic compounds and sensory profiles in cacao nibs was defined with a Pearson simple correlation coefficient using Pandas’s corr function on Python 3.13.2, while the heat map was created using the Seaborn Library. Both procedures were developed in googlecolab, 1.2.0.

To determine the significant effects of the temperature, variety, and its interaction on the concentrations of phenolic compounds a completely random design with factorial arrangement with the proc glm procedure was used; when significant effects were found, the Tukey multiple comparison test was used. The analyses of variance were performed with a significance level of 5% in the SAS Enterprise Guide 7.1.

## 3. Results

### 3.1. Method’s Validation

#### 3.1.1. Linearity, LOD, and LOQ

The calibration curves were evaluated after regression analysis, and linearity was estimated using the coefficients of determination (R^2^) for concentrations ranging from 80% to 100% of the working concentration. The data presented in Table 2 suggest that the standard curves were linear; the R^2^ values suggest that the linear models can explain over 99% of the experimental variability, confirming a satisfactory relationship between analyte concentrations and spectrophotometric responses.

Regarding detection and quantification limits, the procedure was sensitive for detecting and quantifying theobromine, caffeine, catechin, epicatechin, epigallocatechin, and total phenols in the cacao samples, with no significant interferences in the instrumental technique. Moreover, regression analyses showed a significant linear correlation between concentration and signal intensity (*p* < 0.05).

#### 3.1.2. Precision

Table 3 shows the precision of each method, which is evaluated as the standard deviation. The coefficient of variation (CV) is considered acceptable as it does not exceed 10% of the target value. This suggests that the methods are precise and indicates.

### 3.2. Method’s Application

#### Fermentation Temperature and Variation of Phenolic Compound Concentration

According to [15], the average fermentation temperatures for the different treatments were as follows: T1 at 41.14 ± 3.84 °C, T2 at 42.43 ± 4.39 °C, and T3 at 43.86 ± 4.74 °C. During the first four days, the temperature increased across all treatments, reflecting the exothermic phase of fermentation, with average temperatures of T1 at 40.00 ± 4.06 °C, T2 at 41.00 ± 4.47 °C, and T3 at 42.6 ± 5.18 °C. During the isothermal phase, from the fourth to the sixth day, temperatures stabilized, maintaining an average of 44.00 ± 0.1 °C for T1, 46.00 ± 0.1 °C for T2, and 47.00 ± 0.1 °C for T3. Thus, at the isothermal phase, pairwise mean t-test differences showed statistically significant differences between T3 vs. T1 [mean difference = 3 (2.654–3.346, 95% CI); t6 = 21.21, *p* < 0.0001], T2 vs. T1 [mean difference = 2 (1.654–2.346, 95% CI); t6 = 14.14, *p* < 0.0001], and T3 vs. T2 [mean difference = 1 (0.654–1.346, 95% CI); t6 = 7.07, *p* = 0.0004].

According to Figure 1, for the CCN 51 variety, the catechin content increases with treatments T1 and T2 until day 5, which begins to decline. In the case of treatment T3, an increase in catechin content is observed until day 2, followed by a decrease in concentration between days 2 and 5 of fermentation. Epicatechin, on the other hand, decreases from the start of fermentation until day 5, then experiences a slight increase until day 6.

The responses to the treatments for the IC95 variety are quite varied. For treatment T1, the concentration of catechins increases throughout the six days of fermentation, particularly for epigallocatechin. In contrast, for treatment T2, the concentration of all catechins decreases on day 2 and then increases on day 5 of fermentation. With treatment T3, the concentration of all catechins gradually decreases throughout the fermentation process.

In the case of the TCS01 variety with treatment T1, there was an approximately 98% decrease in epigallocatechin concentration from the start of fermentation to day 5. At the same time, catechin maintained a constant trend throughout fermentation. With treatment T2, a decrease in all catechins was observed on day 2 of fermentation, followed by a significant increase on day 5. For treatment T3, the trend of all catechins increases until day 6 of fermentation.

This is attributed to the fact that the temperatures of all treatments increased until the fourth day of fermentation, simulating an exothermic phase, and, from the fourth to the sixth day, they were isothermal. In most cases, the concentration of catechins increased from the start of fermentation until day 5, followed by a decrease until day 6.

The main methylxanthines of cacao are theobromine (3.7% on a fat-free basis) and caffeine (about 0.2%). As shown in Figure 2 for methylxanthine content, their behavior is very similar to catechins; a decrease in concentration is observed from the start of fermentation until day 5, followed by an increase until day 6 of fermentation. For the CCN51 variety, theobromine decreases throughout fermentation for treatment T1, while, in treatment T2, it increases starting from day 2. In treatment T3, it increases during the first 48 h and then begins to decrease until day 5 before increasing again during the last 24 h of fermentation.

Caffeine, on the other hand, shows a more stable trend compared to theobromine. For treatments T1 and T3, a decrease is observed throughout fermentation, while, in treatment T2, caffeine content increases until day 5, followed by a decrease.

The ICS95 variety exhibits a more variable trend across all treatments. For treatments T1 and T3, theobromine concentration increases during the first two days of fermentation, followed by a decrease toward day 5. In treatment T2, the trend is the opposite: a decrease on day 2 and an increase on days 5 and 6 of fermentation. Caffeine generally shows a more stable trend, with a slight decrease observed in treatments T2 and T3 throughout fermentation.

For the TCS01 variety, theobromine increases during the first two days of fermentation, followed by a decrease by day 5 in treatments T1 and T3. Caffeine gradually increases until day 5 for treatments T2 and T3, while, in treatment T1, it decreases until day 5 and then increases to a lesser extent during the last 24 h of fermentation until day 6.

The results shown in Figure 3 indicate that, for treatment T1, across the three evaluated varieties, there is a tendency for the concentration to decrease during the first two days of fermentation, followed by a significant increase by day 5. In contrast, treatment T2 shows a more stable trend in the concentration of total polyphenols throughout fermentation for all three varieties. For treatment T3, a decrease is observed during the first five days of fermentation, followed by a slight increase on day 6 for the CCN51 and TCS01 varieties. The correlation between time and temperature fermentation on the phenolic compound concentration in cacao nibs is shown in Table 4.

Table 4 reports the relationships for the average phenolic compound content determined in cacao nib samples. It was observed that the concentration of theobromine increased with time during fermentation (6 days) with a *p*-value 0.011. As for the temperature variable, the T2 treatment, with an average temperature of 42.43 °C, has the highest values of theobromine, caffeine, catechin, epicatechin, and epigallocatechin. In general terms, temperature was a variable that affected the content of phenolic compounds. Moreover, the correlation between time and temperature was evaluated, and the results showed only a significant correlation with theobromine with a *p*-value of 0.0383. 

The results of the sensory analysis in Figure 4 show that the predominant flavors in the samples evaluated under the three treatments were Bitter, Astringent, Acidic, Cocoa, Fruity, and Citrus. According to the genotype, significant differences were found in the correlations (Figure 5). For CCN51, a low direct correlation was observed between the concentration of epigallocatechin and the bitter, astringent, and acidic tastes. The concentration of total phenols showed a weaker positive correlation with astringent, sour, and cocoa flavors and a slightly lower correlation with caffeine concentration.

For the ICS95 genotype, low positive correlations were observed between bitter taste and the concentrations of total phenols, caffeine, and epicatechin. Cocoa flavor showed a low but significant positive correlation with the concentrations of epigallocatechin, catechin, caffeine, epicatechin, and total phenols. Citrus flavor was positively and moderately correlated with total phenol concentration and, to a lesser extent, with the concentrations of caffeine and epicatechin.

For the TCS01 genotype, low positive correlations were found between bitter taste and epigallocatechin concentration. Cocoa flavor showed a low positive correlation with the caffeine concentration and, to a lesser extent, with the theobromine and epigallocatechin concentrations. For this genotype, moderate negative correlations were observed between citrus flavor and the concentrations of theobromine and catechin.

The correlation between the phenolic compound concentration and sensory profile is shown in Figure 5.

## 4. Discussion

### 4.1. Effect of Temperature Fermentation on Phenolic Compound Concentration

Temperature plays a crucial role in the fermentation process. If the temperature is too low, it slows down the fermentation and prolongs the overall duration. Increasing the temperature ensures the death of the embryo and triggers numerous biochemical reactions within the cacao bean, leading to the development of aroma and flavor precursors [14,19].

During cacao fermentation, temperature and pH are essential variables to control. An increase in these variables promotes microbial activity through anaerobic processes, converting sugars into ethanol and subsequently producing lactic and acetic acids [18,20]. Temperature also influences enzymatic activity, with higher temperatures accelerating enzymatic reactions, such as the breakdown of pectin and other polysaccharides. This process facilitates the release of compounds that contribute to the cacao’s flavor profile [21].

Catechins, including epicatechin, catechin, epigallocatechin, and procyanidin, are responsible for the bitter and astringent taste of roasted cacao [8]. However, other authors, such as [22], propose that amino acids also contribute to cacao’s bitter flavor. The results showed that, in general, catechin concentrations increased from the beginning of fermentation until day 5, followed by a decrease by day 6. However, catechin exhibited a constant trend throughout the fermentation process. These findings align with those reported by [9], who observed that epicatechin concentrations increase during maturation while catechin levels remain unaffected. The decrease in epicatechin concentration, typically between 10% and 20% during fermentation, is attributed to the diffusion of polyphenols into the exudate, which is subsequently drained from the system [14].

The diffusion of acetic acid, lactic acid, and ethanol into the beans is driven by high temperatures during fermentation and promotes flavonoid degradation, which improves cacao quality by reducing bitterness and astringency [16]. This reduction in bitterness is linked to the decreased catechin content observed in treatments T2 and T3 compared to T1. These results are consistent with the findings by [15] where a sensory analysis demonstrated that higher average temperatures, such as those in T3, contribute to the development of high-quality flavors, particularly in the TCS01 genotype.

Similarly, temperature increases during fermentation are crucial for the reduction in methylxanthine content, such as theobromine and caffeine, which are key contributors to cacao’s bitter taste. The data further indicate that the concentration of methylxanthines exhibited a pattern similar to that of catechins, showing an overall decrease from the onset of fermentation through day 5, followed by a subsequent rise on day 6.

According to [19], theobromine concentration decreases after the first 72 h of fermentation, a trend observed in treatments T1, T2, and T3 across all three genotypes. This reduction in theobromine content is attributed to the exudation of compounds from the beans during fermentation. Similarly, caffeine exhibited a comparable behavior to theobromine in treatments T2 and T3, aligning with the findings reported by [23], which showed that the caffeine content is also significantly reduced during cacao fermentation.

Several studies demonstrate that reducing methylxanthines significantly improves the sensory profile by reducing the bitterness and astringency [2,24].

Finally, the evaluation of the total phenol content determined by UV–Vis analysis revealed a decrease during the first five days of fermentation in treatments T2 and T3, and during the first two days in T1, across all three genotypes evaluated. These findings are consistent with reports by [21,25] who documented a reduction in polyphenols during the fermentation and drying processes. This reduction occurs as polyphenols diffuse from storage cells and undergo oxidation, leading to the formation of high-molecular-weight, insoluble compounds known as tannins. These tannins contribute to the development of cacao’s aromatic profile [14]. This process is driven by both enzymatic and non-enzymatic reactions, with polyphenol oxidase serving as the primary catalyst [21]

### 4.2. Correlation Between Phenolic Compounds Concentration and Sensory Profile

The polyphenol content in cacao varies by genotype, growing conditions, origin, and other factors, with estimates suggesting that cacao beans contain between 1.3% and 2.3% polyphenols [26]. The most common polyphenols in cacao are proanthocyanidins (58–65%), catechins (29–38%), and anthocyanins (4%).

Cacao’s origin and fermentation process significantly influence changes in the concentrations of bioactive compounds [27]. Additionally, genetic variety plays a key role in determining the biochemical composition of cacao beans [14,28].

Polyphenols and alkaloids are the primary contributors to the bitter and astringent taste of unfermented cacao [29]. The polyphenol content in dried cacao beans is estimated to range between 6% and 8% [5]. While a high polyphenol content enhances the astringency appreciated by some consumers for its association with antioxidant capacity, it can also lower cacao’s market quality. According to [21], not all polyphenols contribute equally to astringency; smaller molecules, such as epicatechins and procyanidins with fewer than three units, are responsible for the astringent sensation, while larger, insoluble molecules do not.

The high content of proanthocyanidins (tannins) is attributed to cocoa’s astringency, as salivary proteins and mucopolysaccharides diminish the lubricating properties of saliva and produce the characteristic sensation of dry mouth [30]. The study found moderate positive correlations between astringent taste and the concentrations of epigallocatechin and catechin, with weaker correlations linked to the total phenol content, where tannins are present. Catechin, a component of cacao tannins, also influences the beans’ color and astringency [12,31].

Bitterness in cacao is predominantly attributed to alkaloids such as theobromine, caffeine, and theophylline [12,32]. Studies have demonstrated correlations between bitterness and catechin concentrations (e.g., epigallocatechin and epicatechin), with weaker correlations to theobromine and caffeine.

The fermentation process plays a vital role in improving cacao’s sensory profile by reducing theobromine and caffeine levels, which, in turn, decreases bitterness and astringency. These alkaloids are also bioactive, promoting serotonin release in the brain and aiding bronchial relaxation [33]. Controlling methylxanthine concentrations during fermentation is crucial to enhancing palatability while maintaining bioactive properties, ultimately influencing the quality of the final chocolate product [2]. Additionally, fruity and nutty flavors were identified in cacao beans. Nutty notes are likely due to the presence of pyrazines, although alcohols may also contribute to these sensory attributes [14].

## 5. Conclusions

The analytical procedure demonstrated high sensitivity and reliability in detecting and quantifying theobromine, caffeine, catechin, epicatechin, epigallocatechin, and total phenols in cocoa samples. The absence of significant interferences highlights the robustness of the instrumental technique. The strong linear correlation (α < 0.05) between concentration and signal intensity confirms the method’s precision. The acceptable coefficient of variation (CV ≤ 10%) further validates the measurements’ accuracy and reproducibility, reinforcing the analytical approach’s effectiveness.

The presence of methylxanthines and polyphenolic compounds in cacao beans influences their aroma and taste, contributing to bitterness, astringency, and, to a lesser extent, acidity, as well as sour, green, and fruity smells. Correlations between phenolic compounds and sensory attributes indicate that temperature-controlled fermentation can be a valuable tool for improving flavor quality. Optimizing fermentation protocols based on targeted flavor profiles could enhance the market value of cacao products.

However, the study’s genotypic representation was limited to three cacao genotypes (CCN 51, ICS 95, and TCS 01), making the findings less generalizable to other varieties. Additionally, temperature-controlled fermentation does not fully replicate spontaneous fermentation, but identifying key variables provides insights for improving post-harvest processes and ensuring consistent product quality.

A thorough sensory analysis helped pinpoint critical flavor attributes crucial for understanding consumer preferences a key consideration for commercial applications. Future research should quantify polyphenols such as catechin in bean exudates to clarify their behavior and explore possible genotypic variations. Moreover, further studies on microbial dynamics, additional chemical compounds, extended fermentation durations, and consumer perception will be essential to fully understand the role of phenolic compounds in flavor development.

## Figures and Tables

**Figure 1 foods-14-01441-f001:**
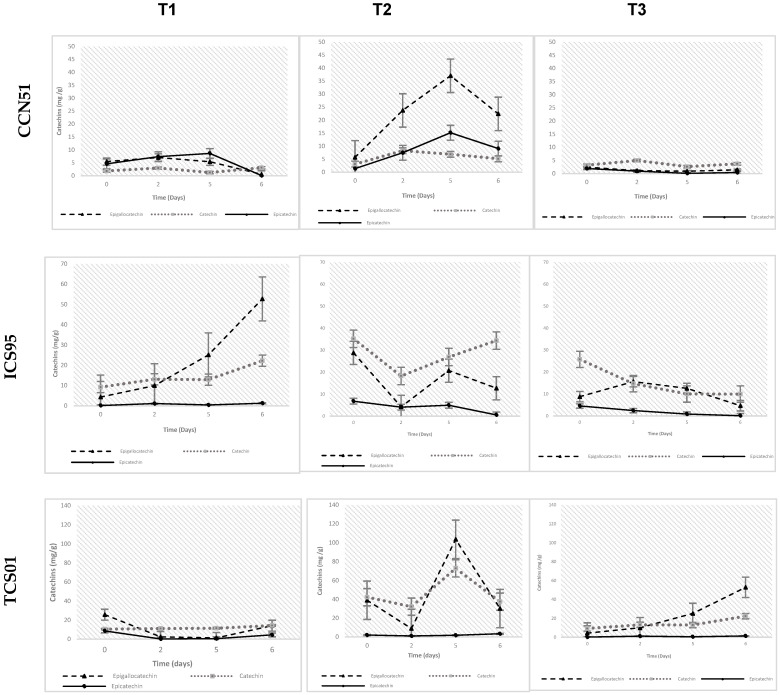
Variations of catechins content (mean ± S.E.) by UPLC-DAD-RI in cacao nibs for three genotypes (CCN 51, ICS 95, and TCS 01) fermented for three temperature profiles (T3, T1, and T2) during 6 days. Own source.

**Figure 2 foods-14-01441-f002:**
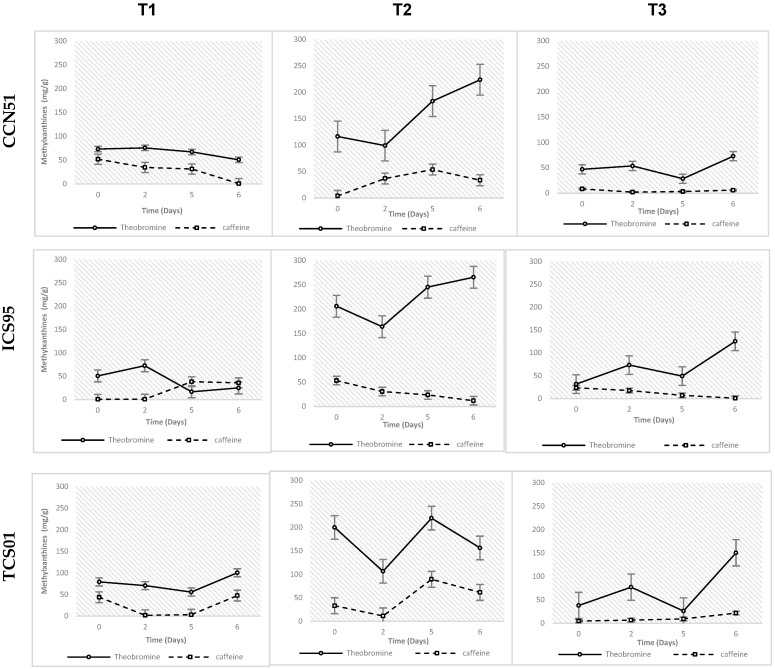
Variations of methylxanthine content (mean ± S.E.) by UPLC-DAD-RI in cacao nibs for three genotypes (CCN 51, ICS 95, and TCS 01) fermented under three temperature profiles (T3, T1, and T2) for 6 days. Own source.

**Figure 3 foods-14-01441-f003:**
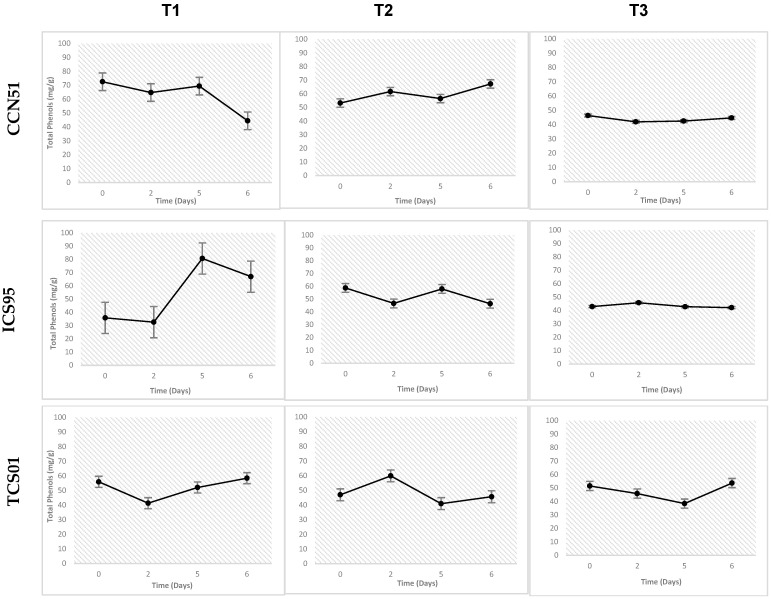
Variations of total phenols content (mean ± S.E.) by UV–Vis in cacao nibs for three genotypes (CCN 51, ICS 95, and TCS 01) fermented under three temperature profiles (T3, T1, and T2) for 6 days. Own source.

**Figure 4 foods-14-01441-f004:**
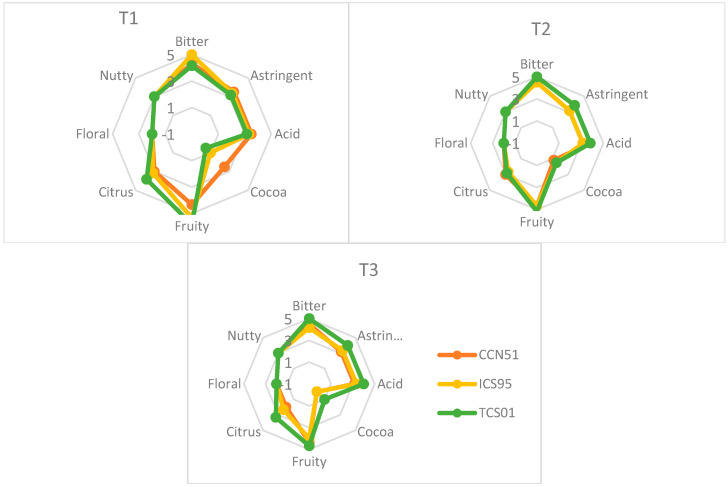
Sensory profiles of cacao liquor for three cacao genotypes (CCN 51, ICS 95, and TCS 01) fermented under three temperature profiles (T3, T2, and T1) [15].

**Figure 5 foods-14-01441-f005:**
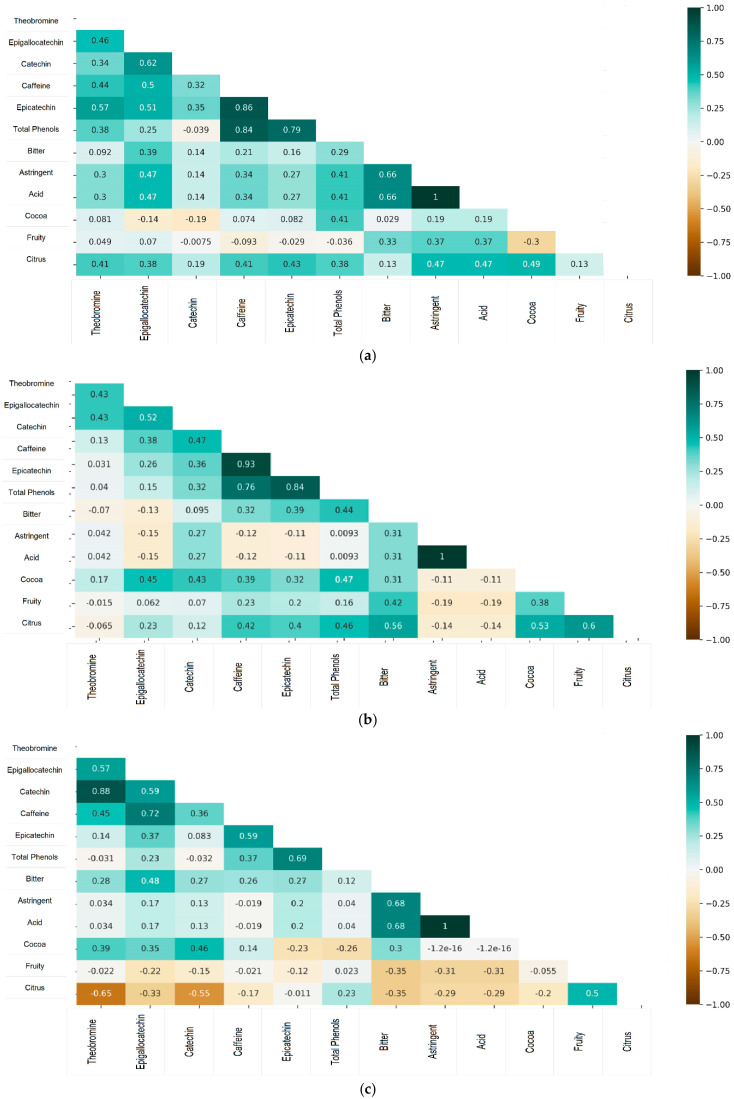
Correlation heat map (Pearson correlation coefficients) with a significance level of 5% among phenolic compounds and sensory variables, for three cacao genotypes: (**a**) CCN 51, (**b**) ICS 95, and (**c**) TCS.

**Table 1 foods-14-01441-t001:** Controlled temperature (T) profiles (°C) for 6 days (144 h) of fermentation. Data were adapted from [15].

Hour	T3	T1	T2
0	35	35	35
24	40	37	38
48	44	40	42
72	46	44	44
96	48	44	46
120	47	44	46
144	47	44	46

**Table 2 foods-14-01441-t002:** Linearity, LOD, and LOQ for UPLC-DAD-RI and UV–Vis methods.

Phenolic Compound	Theobromine	Caffeine	Catechin	Epicatechin	Epigallocatechin	Total Phenols
Standard	Theobromine	Caffeine	(+)-catechin	(-)-epicatechin	(-)-epigallocatechin	Tannic Acid
Working range (ug-mL^−1^)	0–50	0–50	0–50	0–50	0–50	0–100
LOD (ug-mL^−1^)	0.0031	0.0011	0.0012	0.0006	0.0147	1.83
LOQ (ug-mL^−1^)	0.0105	0.0037	0.0039	0.0020	0.0491	6.09
Regression equation	y = mx + b	y = mx + b	y = mx + b	y = mx + b	y = mx + b	y = mx + b
Correlation coefficient ^®^	0.9965	0.9998	0.9994	0.9991	0.9969	0.9940
Slope (m)	24,624	53,575	16,674	16,858	787.04	0.0146
Intercept (b)	14,265	−526.4	1717.8	1690.1	382.5	0.3211

**Table 3 foods-14-01441-t003:** Accuracy for UPLC-DAD-RI and UV–Vis methods.

Phenolic Compound	Theobromine	Caffeine	Catechin	Epicatechin	Epigallocatechin	Total Phenols
Standard	Theobromine	Caffeine	(+)-catechin	(-)-epicatechin	(-)-epigallocatechin	Tannic Acid
Intermediate accuracy (RSD) ^a^	2.49	0.72	12.17	0.94	246.3	0.044
CV (%)	4.54	5.40	7.29	5.85	4.05	3.05

^a^ The inter-day relative standard deviation of the concentrations evaluated.

**Table 4 foods-14-01441-t004:** Average concentrations ± standard error in mg g^−1^ of theobromine, caffeine, catechin, epicatechin, epigallocatechin, and total phenols in cacao nibs.

Factor		Theobromine	Caffeine	Catechin	Epicatechin	Epigallocatechin	Total Phenols
Temperature (Treatment)	T1	61.507 ± 11.295 ^a^	24.242 ± 6.433	9.204 ± 2.476	4.240 ± 0.990	6.051 ± 3.749	56.247 ± 3.197
	T2	182.173 ± 11.295 ^b^	36.960 ± 6.433	26.879 ± 2.476	4.794 ± 0.990	28.064 ± 3.749	53.493 ± 3.197
	T3	64.505 ± 11.295 ^c^	9.317 ± 6.433	11.023 ± 2.476	1.177 ± 0.990	11.620 ± 3.749	44.862 ± 3.197
	*p*-value	<0.0001	0.013	<0.0001	0.025	<0.0001	0.037

Means for each variable with a different letter for the day of fermentation or treatment indicate significant differences (*p* = 0.05).

## Data Availability

The original contributions presented in the study are included in the article/Appendix A, further inquiries can be directed to the corresponding author.

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
