# Peer review of "The Impact of Controlled Fermentation Temperature on Chemical Composition and Sensory Properties of Cacao"

_foods, 2025, doi:10.3390/foods14091441_

Round 1
Reviewer 1 Report
Comments and Suggestions for Authors
In this work entitled, “Fermentation temperature and its effect on phenolic com-2 pounds concentration in cocoa beans”, Calvo et al., attempted to develop an UPLC-MS/MS-based analytical approach for quantitation of the major secondary metabolites found in cocoa beans. The authors also evaluated the impact of temperature treatment on the phenolic and alkaloids content of the fermented beans. In all, it is a well-designed study with some interesting findings. Nonetheless, there are still some issues that warrants clarification prior to further consideration of the submission. Authors are encouraged to carefully address the concerns indicated in the comments below.
-Lines 22-23: “The average temperature throughout fermentation of the 22 different treatments was T1: 41.14 ± 3.84 °C, T2: 42.43 ± 4.39 °C, and T3: 43.86 ± 4.74 °C.” These temperature treatment appear to be quite similar. Are they significantly different? And what informed the authors’ choice of these specific temperatures for the treatment?
-The Abstract is quite lengthy. Authors should try to be more concise. Secondly, the Abstract is replete with qualitative phrases but without quantitative values to support them. Please include some quantitative numerical data to support your principal findings.
-Please verify the citation style and use that which is recommended by the journal guideline for authors.
-Line 66-67: “Fermentation significantly affects flavonoid content, as flavonoids contribute to bean astringency and color and are ethanol-soluble” Rewrite this sentence to improve the clarity, perhaps it should be separated into two phrases.
-Line 71: ‘cacao’ or cocoa? Although both mean the same thing, please always ensure that you are consistent in your writing.
-Lines 82-83: “For example, an isothermal range of 45–50°C promotes flavonoid degradation, thereby 82 reducing bitterness and astringency”. A few lines prior (70-71) the authors alluded that the bitter taste was due to the presence of alkaloids (e.g., theobromine, etc.). Moreover, not all flavonoids produce a bitter taste. Please be more specific.
-Lines 84-90. The rationale for this study is not clear. Authors should improve this section with better articulation of the research hypothesis and gap.
- Line 159: “2.4. Cocoa phenol extraction and analysis”. This sentence should rather be changed to ‘Extraction and analysis of cocoa phenolic-rich extract.’
-Line 173-177: This section should be improved. ‘UPLC’ should be written in full (since this is the first time it is being mention) and this is apparently UPLC-MS. Please correct it.
-In the discussion section, no clear explanation is provided for the reversal of phenolic and alkaloids content with progression of time. Please correct this deficiency.
-The conclusions section should be improved. The claim that “The present study showed that increasing the average temperature during fermentation is decisive for reducing the concentration of catechins and methylxanthines, considerably improving the quality of cocoa in sensory terms.” Is questionable given that the temperatures treatments do not seem to be statistically different.
-Authors should also clearly explain the limitations of the current study and the specific implications of the findings in Foods.
-Considering that a substantial part of this work dealt with the development of UPLC-MS approach for the analysis of the compounds, the authors should perhaps comment on this in the conclusions.
Author Response
Comment has been attached.

Reviewer 2 Report
Comments and Suggestions for Authors
This study aimed to explore the effect of temperature-controlled fermentation on the concentration of phenolic compounds and methylxanthines in cocoa beans, as well as their sensory profile. The paper needs to be revised. A few suggestions are provided below:
- The title of the manuscript needs to be modified, as its current form does not reflect the results of the sensory analysis, which is an important part of the study and is discussed in detail. Proposed title: "The Impact of Fermentation Temperature on Chemical Composition and Sensory Properties of Cocoa"
- The manuscript does not describe the research objective. Please add the research objective at the end of the Introduction section.
- I suggest removing section “2.6. Method’s Application” and integrating the content of this subsection into section 2.4. Cocoa phenol extraction and analysis.
- Please complete the "Sensory analysis" section with a brief description of the research methodology. The reference to the previous work, which contains a detailed description of the study, is appropriate. However, adding a concise summary of the methodology in this paper will make it easier for the reader to analyze the results. Are the presented sensory analysis results the same as those already described in the previous study (Camargo, I. D., Rodriguez-Silva, L. G., Carreño-Olejua, R., Montenegro, A. C., & Quintana-Fuentes, L. F, 2024)? If so, the reference should be included in the description of Figure 4. The quality of Figure 4 needs improvement.
- Statistical analysis section requires improvement. Proc glm is a procedure in SAS for analyzing variance (ANOVA), covariance (ANCOVA), regression, and more. Please describe the statistical analysis method used to analyze the obtained results (Pearson correlation coefficient).
- “UV-VIS”, „UV/VIS” - Please standardize it.
- The * symbols in the entire Table 1 are unnecessary. The reference to the previous work in the table title is sufficient.
- Line 255 - The sentence is incomplete.
- Section 3.2 should be reworded so that the previous work is not copied (quoted) directly.
- Table 4 needs improvement; its description and content are unclear. According to the description, it contains the average contents of individual polyphenolic compounds, but the table includes results for substances like theobromine and caffeine. Why are only the results for theobromine presented (the effect of fermentation time on chemical composition)? However, the description states that Table 4 reports the relationships for the average phenolic compound content determined in cacao nibs samples. This section should be revised to eliminate any ambiguities.
- In Table 2, Table 4 please replace the commas with periods.
- The authors should revise the Conclusions and include a discussion of the limitations of their study and provide more detail on the novelty of their research.
The manuscript requires language editing.
Author Response
Comments are shown in the attached file

Reviewer 3 Report
Comments and Suggestions for Authors
The quality of cocoa beans is significantly influenced by the raw materials and processing techniques employed. This manuscript investigates the impact of three temperature regimes on the phenolic content and sensory attributes of three cocoa bean genotypes during fermentation. The study reveals that fermentation temperature significantly affects phenolic content, and increasing fermentation temperature can enhance the sensory quality of cocoa. This research contributes to the production of high-quality cocoa bean products. While the experimental design appears sound and the research is relatively comprehensive, there are deficiencies in the presentation and analysis of the results. The following comments are provided:
- The x-axis in Figures 1-3 should utilize non-uniform intervals, as the fermentation time points (0, 2, 5, and 6 days) do not constitute an arithmetic sequence. Alternatively, consider using bar graphs instead of line graphs. Additionally, the rationale for measuring parameters on day 5 rather than day 4 should be clarified.
- At 0 days of fermentation in Figures 1-3, the phenolic content should be consistent across different temperature treatments for the same cocoa bean variety. However, the figures show significant discrepancies at 0 days, such as in the caffeine content of CCN51, the epigallocatechin content of ICS95, and the total phenolic content across all varieties. The authors need to address these inconsistencies.
- In Figure 3, the total phenolic content for TCS01 fermented for 5 days under the T2 treatment is reported as 40 mg/g, whereas Figure 1 indicates that the combined content of epigallocatechin and catechin alone reaches 160 mg/g. Similar large discrepancies in content are observed in several other instances.
- The rationale for the inclusion of lines 334-335 and line 345 should be provided.
- It is recommended to include photographs of the cocoa beans under each treatment.
- Figure 4 requires improvement, as its readability is poor.
- The significance of the correlation coefficients in Figure 5 needs to be indicated.
- The absence of "Nutty" and "Floral" in the correlation analysis should be explained.
- The conclusions drawn in lines 476-479 are not adequately supported by the correlation analysis results.
- The formatting of the references is inconsistent and requires careful revision.
Author Response
Comments are shown in the attached file

Round 2
Reviewer 1 Report
Comments and Suggestions for Authors
The changes made are acceptable.